# Incidental Diagnosis of Rheumatic Myocarditis during Cardiac Surgery—Impact on Late Prognosis

**DOI:** 10.3390/diagnostics13203252

**Published:** 2023-10-19

**Authors:** Paulo Pinto Alves Campos Vieira, Rodrigo Furtado Pereira, Carlos Eduardo Barros Branco, Vitor Emer Egypto Rosa, Marcelo Luiz Campos Vieira, Lea Maria Macruz Ferreira Demarchi, Livia Santos Silva, Luiza Guilherme, Flavio Tarasoutchi, Roney Orismar Sampaio

**Affiliations:** 1Campus 1 Interlagos, Santo Amaro University, Sao Paulo 04829-300, Brazil; paulo.alves.vieira22@gmail.com; 2Clinical Hospital (HC), São Paulo University Medical School, Sao Paulo 05403-000, Brazil; rodrigo.furtado@fm.usp.br; 3Heart Institute (InCor), São Paulo University Medical School, Sao Paulo 05403-000, Brazil; carlosbbranco@hotmail.com (C.E.B.B.); mluiz766@terra.com.br (M.L.C.V.); leademarchi@uol.com.br (L.M.M.F.D.); luizagui@usp.br (L.G.); tarasout@uol.com.br (F.T.); 4Clinical Hospital, Santa Marcelina University, Sao Paulo 08270-140, Brazil; lsantoss0412@gmail.com

**Keywords:** rheumatic fever, diagnosis, Aschoff body, pathology, surgery

## Abstract

Rheumatic fever (RF) and rheumatic heart disease (RHD) are still highly prevalent, particularly in low- and middle-income countries. RHD is a neglected and underdiagnosed disease for which no specific laboratory diagnostic test is completely reliable. This is a retrospective observational study, which included 118 patients with RHD who underwent cardiac surgery from 1985 to 2018. The aim of this investigation was to evaluate the clinical, epidemiological, echocardiographic and pathological characteristics in two cohorts of RHD patients: one cohort with Aschoff bodies present in their pathological results and the other without such histopathological characteristics. No conventional clinical and laboratory tests for RHD myocarditis were able to identify active carditis during the preoperative phase of valve repair or replacement. Patients who had Aschoff bodies in their pathological results were younger (median age of 13 years (11–24 years) vs. 27 years (17–37 years), *p* = 0.001) and had higher rate of late mortality (22.9% vs. 5.4%, *p* = 0.043). In conclusion, the presence of Aschoff bodies in pathological findings may predict increased long-term mortality, emphasizing the importance of comprehensive pathology analysis for suspected myocarditis during heart surgery.

## 1. Introduction

Rheumatic fever (RF) and rheumatic heart disease (RHD) are consequences of untreated or improperly managed infections due to group A beta-hemolytic streptococcus. After an episode of acute rheumatic fever, approximately 30–45% of individuals are at risk of developing RHD [1]. This condition contributes significantly to morbidity and mortality, particularly in low- and middle-income countries, thereby imposing a substantial burden upon healthcare systems [2,3]. Recurrent episodes of rheumatic myocarditis increase valvular deterioration, leading affected patients towards progressive disease and eventual cardiac surgery. Notably, the diagnosis of rheumatic myocarditis is often difficult and misleading, especially in patients with severe concomitant valvular heart disease [4].

No single clinical or laboratory finding can definitively diagnose acute rheumatic fever (ARF), except possibly in the case of Sydenham chorea. Consequently, a set of diagnostic criteria was initially formulated in 1944 by Jones and his colleagues. Subsequently, these criteria underwent four revisions by both the World Health Organization (WHO) and the American Heart Association (AHA) between 1965 and 2003 [5]. These criteria have become increasingly specific while reducing sensitivity, a modification undertaken to align with the condition’s infrequent occurrence in higher-income nations. More recently, efforts have been made to increase sensitivity and specificity in areas of higher prevalence of RHD and RF [6].

Since its initial delineation, the Aschoff body has been universally acknowledged as the histological cornerstone of RHD and a distinctive marker denoting the presence of acute or active rheumatic disease. These characteristic structures can manifest within various cardiac tissues, with a predilection for the myocardial perivascular connective tissue. They are more prevalent within the valvular tissue of individuals affected by RHD [7,8].

Histologically, an Aschoff body is a granulomatous lesion composed of collagen-degenerative alterations surrounded by multinucleated histiocytes called Aschoff cells, in which some of these cells have a caterpillar appearance (Anitschkow cells) and a variable number of neutrophils, eosinophils, plasmacytes and lymphocytes [7,8].

An Aschoff body presents with three morphological phases of development (Figure 1). The early or exudative phase is characterized by fibrinoid necrosis or degenerative fibroblast alterations, and inflammatory cells such as neutrophils, eosinophils, lymphocytes and macrophages, including Aschoff and Anitschkow cells [8,9].

The proliferative phase (Figure 1) comprises the tissue damage repairing process and is when granulation tissue and inflammatory cell infiltrates containing lymphocytes and macrophages, including Aschoff and Anitschkow cells, develop. The cicatricial phase is characterized by the gradual replacement of the granulation tissue and inflammatory infiltration with scar tissue [8,9].

Verrucae (Figure 1) are other morphological, but unspecific, features of RF/rheumatic activity. They are small vegetations located on the occlusion line of cardiac valves in patients with RHD. They are composed due to superficial fibrinoid necrosis and the deposition of fibrin and platelet cells in the valvular tissue, leading to a palisade of lymphomononuclear cell infiltrates [8,9].

Aschoff nodules arise from the aggregation of inflammatory cells, predominantly encompassing lymphocytes and plasma cells [10]. These distinctive structures serve as virtually conclusive evidence of acute rheumatic fever, which may impact whether a patient receives an early or long-term prognosis after valve surgery [11,12]. Consequently, the objective of this study is to evaluate the clinical and echocardiographic characteristics from a cohort of patients in whom Aschoff bodies (AB) were incidentally detected within myocardial and valvular biopsies undertaken during heart valve surgery.

## 2. Methods

### 2.1. Aim of the Study

The aim of this study was to correlate the presence of Aschoff bodies and verrucae found in surgical biopsies after valvular heart disease surgery with preoperative information, described as clinical, laboratory, epidemiological and echocardiographic data, and with long-term clinical postoperative follow-up data.

In this context, we systematically examined the following aspects:Clinical manifestations of rheumatic myocarditis/valvulitis:
Evaluation of indicative clinical manifestations such as fever, tachycardia and progressive cardiac failure.Monitoring the utilization of dobutamine and corticoids for management during the immediate postoperative period.Assessment of the efficacy of secondary prophylaxis with benzathine penicillin.
Laboratory indicators of inflammation and infection:
Analysis of key laboratory markers including C-reactive protein (CRP) levels.Measurement of erythrocyte sedimentation rate (ESR).Leukocytosis to assess inflammatory and/or infectious status.
Clinical follow-up and patient’s characteristics:
Age.Gender.Early in-hospital mortality.Ten-year follow-up after surgery, either through medical records or phone calls.
Echocardiographic parameters:Quantification of left ventricle ejection fraction (LVEF).Identification of specific valvular heart disease disorders.Pathology:
Characterization of pathological findings encompassing the inflammatory stage.Identification and analysis of the presence of Aschoff bodies.


### 2.2. Patients

Our study involved a thorough analysis of a cohort comprising 118 patients, all of whom were suspected of having rheumatic carditis by either a surgeon or a clinician. These patients underwent biopsy procedures during heart valve surgeries, aligning with established international guidelines [13], within the time frame spanning from 1985 to 2018. Our analytical approach drew upon an in-depth review of the hospital’s medical records, complemented by telephone calls with the patients who had undergone the surgical interventions or, if this was not possible, with their relatives.

### 2.3. Histopathological Analysis

Surgical specimens were obtained from 118 patients with RHD who underwent surgical procedures for rheumatic valvular heart disease. Tissue samples were fixed in neutral buffered 10% formalin and embedded in paraffin. They were cut into 4–5 μm histological sections and stained with hematoxylin and eosin. Histological examination was performed by at least two pathologists who assessed the presence of verrucae, Aschoff bodies, inflammatory infiltrates, vascular proliferation, fibrosis and calcification in each cardiac valve and myocardium sample.

Aschoff bodies were described according to their morphological features as exudative, proliferative or cicatricial (Figure 1).

Thus, the patients were divided into 2 subgroups for analysis: 1—the Aschoff Bodies Group (ABG), in which Aschoff bodies or verrucae suggestive of rheumatic carditis were found during biopsy or valvular samples obtained during cardiac surgery; and 2—the No Aschoff Bodies Group (NABG), in which there were no pathological findings of Aschoff bodies or verrucae.

### 2.4. Statistical Analysis

Continuous variables are presented as medians (interquartile ranges), while categorical variables are presented as percentages. The Mann–Whitney U-test was applied for continuous variables and the Fisher exact test or χ2 test was applied for categorical variables, as appropriate. A *p*-value < 0.05 was used to indicate statistical significance, and all tests were two-tailed. The SPSS statistical package, version 20 (IBM, Armonk, NY, USA), was used to conduct the entire analysis.

## 3. Results

The overall profile of the entire patient cohort (*n* = 118) revealed a predominantly youthful demographic, with an average age of 23 years. Among the participants, a significant proportion were female, totaling 72 (61%). Aschoff bodies were detected in 73 patients (61%). A substantial majority, comprising 101 individuals (85%), underwent secondary prophylaxis with benzathine penicillin, underscoring this as the prevalent therapeutic approach. Furthermore, 73 patients exhibited advanced heart failure, with 61% classified as Functional Class III or IV according to the New York Heart Association classification (Table 1). There were no symptoms/signs indicating autoimmune diseases.

Pathological examination revealed the presence of distinct phases of rheumatic acute disease, as illustrated in Figure 1. Among the patients, verruca formations were identified in seven cases. An exudative Aschoff body with fibrinoid necrosis and degenerative collagen changes was noted in six individuals. A more advanced and specific pattern of rheumatic inflammation, characterized by proliferative Aschoff bodies and showing granulomatous lesions with prominent Aschoff cells, was observed in 52 patients. Additionally, eight patients displayed cicatricial Aschoff bodies within the myocardium, accompanied by fibrotic changes which were indicative of a late-stage acute rheumatic inflammatory process. In the context of analyzing the inflammatory state through pathological assessment, no statistically significant difference was observed upon comparison between the two groups (*p* = 0.296).

When conducting a comparative analysis of age between the two groups, it becomes evident that the “No Aschoff Bodies” Group (NABG) exhibited a younger median age of 13 years (with a range of 11 to 24 years), which is in contrast to the Aschoff Bodies Group (ABG), which displayed a median age of 27 years (with a range of 17 to 37 years). In the pre-surgical clinical assessment, tachycardia was noted in 38 patients (32%), with a notably higher prevalence observed in the NABG (53.1%) compared to 30.4% in the ABG. For additional clinical indicators of rheumatic carditis, such as elevated levels of the C-reactive protein (CRP) or erythrocyte sedimentation rate (ESR) prior to surgical intervention, leukocytosis, and an elongated PR interval, no significant statistical variance was observed between the two groups in this regard (Table 1).

Most of patients in both groups underwent elective surgeries (89.9% in the ABG and 67.7% in the NABG) (Table 1).

Excision biopsies of Aschoff bodies were conducted across various sites, with the most prevalent locations being the mitral valve (53.4%) and the myocardium (26%). For patients without detectable Aschoff bodies, their heart biopsies were primarily performed in the mitral valve (50%), followed by a combination of aortic and mitral valve locations (22.5%) (Table 2).

Upon conducting a comparative echocardiographic analysis between both groups, no statistically significant differences were observed when evaluating types of valve diseases (stenosis and regurgitation) or the specific valves affected. The left ventricular ejection fraction (LVEF) was found to be less than 50% in 13.5% of patients within the ABG, while a lower incidence of 2.3% was noted in the NABG (Table 2).

No discernible distinctions were identified in terms of inflammatory responses or the requirement for vasoactive medications during the immediate postoperative phase. However, remarkable variations emerged during the 10-year follow-up period. Mortality rates were notably elevated within the ABG, registering at 22.9%, as compared to a lower rate of 6.7% within the NABG (Table 3).

## 4. Discussion

Our investigation revealed the presence of Aschoff bodies in 61.8% (73 out of 118) of the patients under scrutiny. This finding is in line with earlier studies, which documented the presence of Aschoff bodies in patient cohorts ranging from 16% to 67% [4]. These data underscore the consistent occurrence of Aschoff bodies across various studies and help to enhance our understanding of their prevalence within the context of this condition and their consequences [4].

As observed, preoperative parameters concerning inflammatory state (fever, leukocytosis, CRP levels and erythrocyte sedimentation rate) could not differentiate between the ABG and NABG groups [14]. Interestingly, tachycardia was more common in the NABG, comprising 53.1% of the patient cohort. However, this finding may be of limited clinical significance, requiring the presence of additional markers indicative of an inflammatory state for comprehensive interpretation.

In most of the cases, Aschoff bodies demonstrated an inflammatory active response when biopsied from the mitral valve, and in one third of the cases these responses were observed after myocardial investigation. Despite the use of current technological, updated imaging techniques, myocarditis diagnoses are still challenging [4,15], and many times could just be defined under pathological analyses. Since the first diagnosis criteria described by Jones in 1944, there has been some sort of discrepancy concerning the laboratory criteria and clinical threshold, which requires more attention be paid to the need for and possible evolution of further and future categorization concerning new clinical and laboratory parameters [16]. However, in our sample, the clinical suspicion was low. The main finding of this investigation is the reinforcement of the fact that rheumatic carditis often has few symptoms and histopathological evaluation during cardiac surgery should always be recommended. Additionally, no signs or symptoms of other “autoimmune diseases” besides RHD were found in our data.

Myocarditis denotes an inflammatory cardiac disorder incited by the influence of infectious agents or toxic substances. These agents engender myocyte impairment either through direct mechanisms or by provoking heightened immune system responses. Manifestations may encompass cardiac dilation, a reduction in the ejection fraction and arrhythmic patterns, or they may even precipitate abrupt fatality [16]. The term “acute myocarditis” characterizes the clinical state arising within the preceding 30 days. Contemporary data indicate a tendency for acute myocarditis among young adults, with a notable prevalence among the female demographic [17]. It is well understood that the Aschoff bodies come from and are developed in the myocardium and endocardium structures during the early stages of RHD, evolving to be aggressive toward heart muscle fibers and cardiac valves [18].

The identification of subclinical episodes of rheumatic myocarditis is always a challenge. Considering that this is an immunoinflammatory disease, it is important to identify clinical or laboratory characteristics that could suggest “rheumatic activity” from an early stage. However, as confirmed in this study, only 30% of patients with RHD may have some degree of inflammation or clinical signs of myocarditis. These patients are often referred for surgical treatment without a diagnosis of active myocarditis, which may have implications for an early or late prognosis. Therefore, an active search can provide additional postoperative care options for these individuals.

Although all participants were defined before the COVID-19 pandemic, we would like to reinforce that this disease is a potential differential diagnosis for myocarditis. The SARS-CoV-2 virus instigates humoral dysregulation and inflammatory responses, which may potentially precipitate or exacerbate myocarditis [19,20]. Thus, anyone, including patients with RHD, can become infected by SARS-CoV-2, resulting in potential additional heart damage [21].

Currently, further investigation is warranted, particularly in patients with autoimmune systemic diseases such as RHD, which may have an association with these illnesses [22].

Cardiac magnetic resonance imaging and gallium-67 cardiac scintigraphy may help identify rheumatic myocarditis. In most cases, the initial clinical suspicion is what leads to the performance of these tests. Recently, the use of fluorine-18-fluorodeoxyglucose positron emission tomography (18F-FDG PET/CT) has also been recommended, with good perspectives for clinical use in suspected cases [14,23,24].

The LVEF, a major indicator of cardiac function, exhibited a distinct pattern between the ABG and NABG. Specifically, a higher proportion of patients within the ABG, approximately 13.5%, demonstrated an LVEF below 50%. In stark contrast, a substantially lower incidence of 2.3% was noted in the NABG. The presence of Aschoff bodies, which are characteristic of active rheumatic disease, could potentially contribute to the exacerbation of myocardial dysfunction, leading to a diminished LVEF. These observations align with those in the existing literature, which highlight the role of myocardial inflammation in impairing cardiac contractility and overall function [25].

According to the current data from this investigation, the inflammatory state, as demonstrated by the evidence of Aschoff bodies, apparently might not represent a further burden on surgical patients as in-hospital mortality demonstrated no statistical difference (3% in the ARCG and 0% in the CG). However, a study involving a larger number of patients would be necessary to confirm this hypothesis. In a previous study, surgery during clinical acute rheumatic fever myocarditis had good results, despite concerns about higher morbidity and mortality in those patients [24].

The majority of patients in both groups were on the benzathine penicillin prophylaxis. One hypothesis is that these patients did not properly adhere to this prophylaxis, which is common in long-term scenarios while being usually necessary for RHD [26,27]. We suspected either the quality of prophylaxis may not be ideal, which is relatively common in low- and middle-income countries, or that the treatment might have led to repeated episodes of subclinical myocarditis, even in those patients receiving benzathine penicillin (BZP) regularly, every four months. These subgroups may benefit from additional doses (every three or two weeks) of BZP. Our finding may bring more attention to repeated cases of rheumatic myocarditis, which should be prevented with additional doses of BZP [28]. Not only the first episode, but recurrent carditis can be even more aggressive, which reinforces the importance of antibiotic prophylaxis [29].

This information is highly impactful, considering the epidemiological burden of RHD and the expenditure of the public medical budget in areas at risk for RHD, such as Latin America, Africa and some areas in Oceania, the Middle East and Asia [4,30,31,32].

Our investigation unveiled a distinct divergence in outcomes during the subsequent 10-year follow-up period. Notably, the ABG exhibited markedly elevated mortality rates, with a recorded incidence of 22.9%, in contrast to the significantly lower rate of 6.7% observed within the NABG. This disparity in long-term mortality underscores the potential prognostic significance of Aschoff bodies and their association with adverse clinical scenarios in chronic RHD.

In conclusion, our study demonstrates that the patients in the group in which Aschoff bodies were found were younger and were associated with a higher rate of late mortality. No other specific disparities were observed in terms of the inflammatory response or vasoactive drugs after heart valve surgery. The presence of Aschoff bodies emerges as a potential predictor of higher long-term mortality in patients with RHD. Detailed pathology analysis searching for rheumatic carditis should be performed in suspected cases of myocarditis as seen in heart surgery.

## 5. Limitations

This investigation involved information derived from the analysis of a single study center, which may not represent a broader population of patients from different continents who are affected by rheumatic disease with cardiac involvement. The information derived from hospital records may not contain all data necessary for the most complete clinical and epidemiological characterization of patients. No detailed investigation of other autoimmune diseases was carried out, as there were no clinical signs or symptoms that could suggest another disease but RHD/RF. The histopathological analysis derived from operative fragments may be compromised due to the obtaining of surgical material, which is not always the most reliable method for the representation of myocardial aggression (number of fragments and site of surgical excision).

## Figures and Tables

**Figure 1 diagnostics-13-03252-f001:**
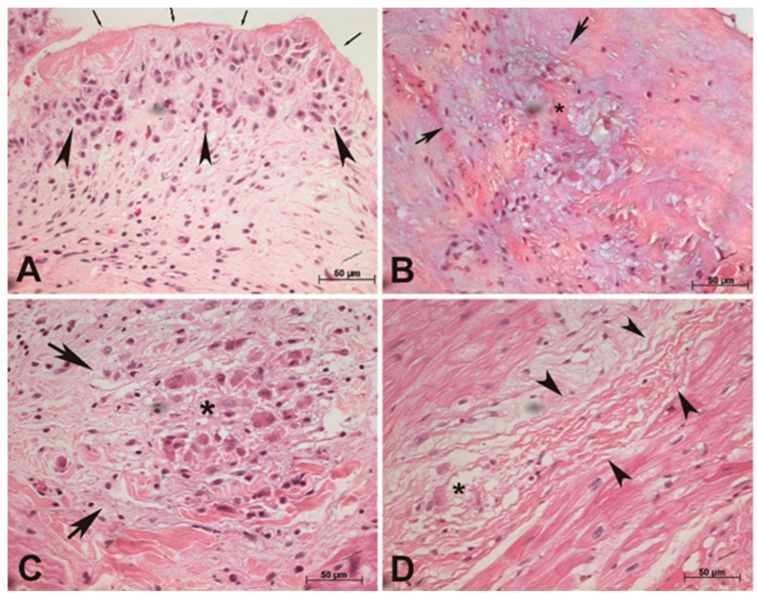
Photomicrographs of histological sections of surgical specimens stained with hematoxylin and eosin. (**A**) ***Verruca* in aortic valve**. Vegetation showing superficial fibrinoid necrosis (arrows) and inflammatory infiltrates with lymphomononuclear cells and eosinophils in the valvular connective tissue (arrowheads); magnification ×400. (**B**) **Exudative Aschoff body in aortic valve**. Edema, eosinophilic degeneration (*) and collagen fragmentation in the valvular tissue without Aschoff cells; magnification ×400. (**C**) **Proliferative Aschoff body in the myocardium of right atrium**. Granulomatous lesion (arrows) with multinucleate and large Aschoff cells and sparse lymphocytes in the interstitial connective tissue with degenerative changes (*); magnification ×400. (**D**) **Cicatricial Aschoff body in the myocardium of left atrial appendage**. The lesion is elongated and has few Aschoff cells (*) and increasing fibrosis (arrowheads); magnification ×400.

**Table 1 diagnostics-13-03252-t001:** Baseline characteristics of patients in study in the Aschoff Bodies Group (ABG) and No ABG.

Baseline Characteristics	*n* = 118	ABG (*n* = 73)	NABG (*n* = 45)	*p*
Age in years	23 (13–35)	27 (17–37)	13 (11–24)	0.001
More than 40 years %	18.6	22.9	10.5	0.188
Female sex %	61	63.5	56.8	0.599
Penicillin prophylaxis %	85	74.2	91.4	0.072
Aschoff bodies %	61.8	100	0	0.0001
Elective surgery %	81.4	89.9	67.7	0.015
Tachycardia %	39	30.4	53.1	0.049
High levels of CRP or ESR %	27.5	31.7	20.8	0.093
Leukocytosis %	27.1	18.3	41.7	0.093
Increase in PR interval %	13	11.5	15.4	1
Signs of cardiac failure %	61	71.2	78.8	0.572
LVEF < 50%	9.3	13.5	2.3	0.006
● Mitral valve %	62.2	55.3	73.5	0.293
● Aortic valve %	6.9	12.4	8.8	0.293
● Combined mitral + aortic valves %	26	31.3	17.6	0.293

ABG—Aschoff Bodies Group; NABG—No Aschoff Bodies Group; penicillin prophylaxis—If have used benzathine penicillin for rheumatic carditis; elective surgery—patients that did not need emergency surgeries; tachycardia—patients with a cardiac frequency over 100 bpm; high levels of CRP or ERS—patients with a CRP > 1.0 mg/dL or erythrocyte sedimentation rate > 22 mm/h; leukocytosis—patients with leukocytes > 10.000 cells/mm^3^; increase in PR interval—considered when electrocardiogram (ECG) results are >200 ms; LVEF—left ventricle ejection fraction; ●—valve disease stenosis or/and regurgitation of moderated or severe grade in the echocardiogram evaluation.

**Table 2 diagnostics-13-03252-t002:** Pathology and follow-up after surgery in the Aschoff Bodies Group (ABG) and No ABG (NABG).

Pathology and Follow-Up Data	ABG (*n* = 73)	NABG (*n* = 45)	*p*
Hospitalization days	17 (14–29)	24 (15–35)	0.074
● Aortic valve %	8.2	15	0.011
● Aortic + mitral valve %	4.1	22.5	0.011
● Mitral valve %	53.4	50	0.011
● Myocardium %	26	12.5	0.011
● Myocardium + mitral Valve %	5.5	0	0.011
ŦProliferative %	50.7	0	0.296
ŦExudative %	8.2	0	0.296
ŦCicatricial %	11	100	0.296
ŦGranulomatous %	4.1	0	0.296
ŦVerruca %	9.6	0	0.296
ŦProliferative + exudative %	8.2	0	0.296
ŦProliferative + verruca %	5.5	0	0.296
ŦProliferative + cicatricial %	2.7	0	0.296
In-hospital mortality %	3	0	0.532

ABG—Aschoff Bodies Group; NABG—No Aschoff Bodies Group; ●—location of excisional biopsy; Ŧ—histological inflammatory stage of the Aschoff body.

**Table 3 diagnostics-13-03252-t003:** Clinical and laboratory parameters from follow-up after surgery.

Post-Procedure	ABG (*n* = 73)	NABG (*n* = 45)	*p*
High levels of CRP or ERS %	36.2	14.3	0.211
Leukocytosis %	14.9	28.6	0.211
Fever %	26.9	29.4	0.972
Corticosteroids %	50	38.2	0.368
Dobutamine %	67.2	64.7	0.981
Heart surgery %	58.7	68.2	0.597
10-Year follow-up mortality %	22.9	5,4	0.043

ABG—Aschoff Bodies Group; NABG—No Aschoff Bodies Group; high levels of CRP or ERS—patients with a CRP > 1.0 mg/dL or erythrocyte sedimentation rate > 22 mm/h; leukocytosis—patients with leukocytes > 10.000 cells/mm^3^; fever temperature > 37.8 °C; corticosteroids—patients that were medicated with corticosteroids.

## Data Availability

The data presented in this study are available upon request from the corresponding author.

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
