# Peer review of "Incidental Diagnosis of Rheumatic Myocarditis during Cardiac Surgery—Impact on Late Prognosis"

_diagnostics, 2023, doi:10.3390/diagnostics13203252_

Round 1
Reviewer 1 Report
An interesting and informative manuscript that has clinical merit; however, there are editing issues that the authors should consider and address. The following are suggestions/comments regarding those issues. Line 33, "... were able to identify active carditis during ...". Line 48, "Rheumatic fever (RF) and ...". Lines 73 & 74, "... number of neutrophils, eosinophils, plasmacytes and ...". Line 77, " ... inflammatory cells such as neutrophils, ...". Lines 102 & 103, "... features of RF/rheumatic activity." Line 104, "... in patients with RHD. They are composed ...". Lines 153 & 154, "... underwent surgical procedures for rheumatic valvular ...". Line 264, "threshold, which brings strong attention ...". Lines 277 & 278, "... early stages in RHD, evolving to the ...". Line 303, "... surgical patients as in-hospital mortality with no statistical ...". Line 305, "In a previous study, surgery ...". Line 321, "... in chronic RHD." Lines 327 & 328, "... of myocarditis seen in heart surgery."
The manuscript is well written.
Author Response
An interesting and informative manuscript that has clinical merit…
R = We thank the Reviewer for his/her comments and for the constructive suggestions that have contributed to improving the manuscript
However, there are editing issues that the authors should consider and address. The following are suggestions/comments regarding those issues.
- Line 33, "... were able to identify active carditis during ...".
- Line 48, "Rheumatic fever (RF) and ...".
- Lines 73 & 74, "... number of neutrophils, eosinophils, plasmacytes and ...".
- Line 77, " ... inflammatory cells such as neutrophils, ...".
- Lines 102 & 103, "... features of RF/rheumatic activity."
- Line 104, "... in patients with RHD. They are composed ...".
- Lines 153 & 154, "... underwent surgical procedures for rheumatic valvular ...".
- Line 264, "threshold, which brings strong attention ...".
- Lines 277 & 278, "... early stages in RHD, evolving to the ...".
- Line 303, "... surgical patients as in-hospital mortality with no statistical ...".
- Line 305, "In a previous study, surgery ...".
- Line 321, "... in chronic RHD."
- Lines 327 & 328, "... of myocarditis seen in heart surgery."
R = We thank the Reviewer for the suggestions. We have corrected this information throughout the manuscript.
Reviewer 2 Report
The results of this paper by Vireira et al., titled “Incidental diagnosis of rheumatic myocarditis during cardiac surgery – impact on late prognosis” led Authors to conclude that the presence of Aschoff bodies in myocardial and valvular biopsies may predict increased long-term mortality. Although patients were recruited before Covid-19 pandemia, in the Discussion the possibility that Covid-19 could contribute to cause myocarditis as a consequance of a derangement in immune system response (Buonacera et al., Int J Mol Sci 2022; Regolo et al., J Clin Med 2022; Regolo et al., J Clin Med 2023) should be brought to the fore. In fact, recently it has been emphasized that Covid-19 pandemic may have a deleterious impact on patients with autoimmune systemic diseases (Ferri et al., Curr Pharm Des 2021; Ferri et al., Lancet Rheumatol 2021). On the other hand, autoimmune diseases are frequently associated with valvular abnormalities (Colaci et al., J Clin Rheumatol 2022), so paving the way to the possible occurrence of incidental "rheumatic myocarditis". The majority of patients had secondary prophylaxis with Benzathine Penicillin, therefore further suggesting a pathogenic mechanism different from classical rheumatic disease.
Specific Comments
1. Were patients carefully screened for autoimmune diseases? In affirmative, please highlight this information in the Results and Discussion.
2. Did patients show symptoms/signs of autoimmune diseases? Please highlight this information in the Results and Discussion.

Author Response
Response to Reviewer #2:
We thank the Reviewer for his/her comments and for the constructive suggestions that have contributed to improving the manuscript.
The results of this paper by Vieira et al., titled “Incidental diagnosis of rheumatic myocarditis during cardiac surgery – impact on late prognosis” led Authors to conclude that the presence of Aschoff bodies in myocardial and valvular biopsies may predict increased long-term mortality. Although patients were recruited before Covid-19 pandemia, in the Discussion the possibility that Covid-19 could contribute to cause myocarditis as a consequance of a derangement in immune system response (Buonacera et al., Int J Mol Sci 2022; Regolo et al., J Clin Med 2022; Regolo et al., J Clin Med 2023) should be brought to the fore.
R = The Reviewer addressed an important point. However, we would like to remember that all data showed in the manuscript were collected before the Covid-19 pandemic.
We added the following paragraph in the discussion and incorporate the suggested references:
“Although all participants were defined before the Covid-19 pandemic, we would like to reinforce that such disease is a potential differential diagnosis for myocarditis. The Sars-Cov-2 virus instigates humoral dysregulation and inflammatory responses, which may potentially precipitate or exacerbate myocarditis” (REF 1 and 2 above). Thus, anyone, including patients with RHD can become infected by Sars-Cov-2, resulting in potential additional heart damage (REF 3 above).
1 - Buonacera A, Stancanelli B, Colaci M, Malatino L. Neutrophil to Lymphocyte Ratio: An Emerging Marker of the Relationships between the Immune System and Diseases. Int J Mol Sci. 2022 Mar 26;23(7):3636. doi: 10.3390/ijms23073636. PMID: 35408994; PMCID: PMC8998851.
2 - Regolo, M.; Sorce, A.; Vaccaro, M.; Colaci, M.; Stancanelli, B.; Natoli, G.; Motta, M.; Isaia, I.; Castelletti, F.; Giangreco, F.; et al. Assessing Humoral Immuno-Inflammatory Pathways Associated with Respiratory Failure in COVID-19 Patients. J. Clin. Med. 2023, 12, 4057.
3 - S. Shi, M. Qin, B. Shen et al., “Association of cardiac injury with mortality in hospitalized patients with COVID-19 in Wuhan, China,” JAMA Cardiology, vol. 5, no. 7, pp. 802–810, 2020.
In fact, recently it has been emphasized that Covid-19 pandemic may have a deleterious impact on patients with autoimmune systemic diseases (Ferri et al., Curr Pharm Des 2021; Ferri et al., Lancet Rheumatol 2021). On the other hand, autoimmune diseases are frequently associated with valvular abnormalities (Colaci et al., J Clin Rheumatol 2022), so paving the way to the possible occurrence of incidental "rheumatic myocarditis".
R = This is an important issue raised by the Reviewer.
We added the following paragraph:
“Currently, further investigation is always warranted, particularly in patients with autoimmune systemic diseases, such as rheumatic heart disease (RHD), where it may have an association with both diseases”.
- Ferri C, Raimondo V, Gragnani L, et al. Prevalence and Death Rate of COVID-19 in Autoimmune Systemic Diseases in the First Three Pandemic Waves. Relationship with Disease Subgroups and Ongoing Therapies. Curr Pharm Des. 2022;28(24):2022-2028. doi:10.2174/1381612828666220614151732
The majority of patients had secondary prophylaxis with Benzathine Penicillin, therefore further suggesting a pathogenic mechanism different from classical rheumatic disease.
R = Nice comment. We suspected either the quality of prophylaxis may not be ideal, which is relatively common in low- and middle income countries or it may have some episodes of myocarditis even in those patients receiving BZP regularly every four months. Those subgroups may benefit with additional doses (every three or two weeks) of Benzathine Penicillin. Our finding may bring more attention to repeated cases of rheumatic myocarditis which should be prevented with additional doses of Benzathine Penicillin.
H C Lue 1, M H Wu, J K Wang, F F Wu, Y N Wu. Three- versus four-week administration of benzathine penicillin G: effects on incidence of streptococcal infections and recurrences of rheumatic fever. Pediatrics Jun;97(6 Pt 2):984-8, 1996
Specific Comments
Were patients carefully screened for autoimmune diseases? In affirmative, please highlight this information in the Results and Discussion.
R = The Reviewer raised an important point. However, all participants have a long clinical history of RHD. So, no specific autoimmune screening was performed once there were no clinical data which indicative of other diseases. The diagnosis of rheumatic fever (RF) was primarily based on the presence of Aschoff's nodules. We add a paragraph in the “limitations” section to address this concern.
Did patients show symptoms/signs of autoimmune diseases? Please highlight this information in the Results and Discussion.
R = No. they did not. We include an additional paragraph in the “results and discussion” to address this concern.
Reviewer 3 Report
The manuscript is engaging and well-written. The English is good. The introduction is complete and valuable to introduce the readers to the study. Still, the purpose of the study should be explained in more detail, trying to emphasize any clinical implications of the results. The tables are well organized, but tables 1 and 2 could be fused into only one table. In my opinion, you should elaborate on the conclusions to emphasize the differences in clinical outcomes between the two study groups.
Author Response
Response to Reviewer #3:
The manuscript is engaging and well-written. The English is good. The introduction is complete and valuable to introduce the readers to the study.
R = We thank the Reviewer for his/her comments and for the constructive suggestions that have contributed to improving our work.
Still, the purpose of the study should be explained in more detail, trying to emphasize any clinical implications of the results.
R = The Reviewer raised an important point. We included this paragraph in the discussion:
“In patients who underwent valve surgery with pathological confirmation of Aschoff’s nodules, the therapeutic clinical management demands a more comprehensive and nuanced approach. This is particularly pertinent due to the heightened mortality risk observed within this subgroup, which comprises a relatively younger demographic”
The tables are well organized, but tables 1 and 2 could be fused into only one table.
R = This is an important issue raised by the Reviewer. We have corrected this information throughout the manuscript accordingly.
In my opinion, you should elaborate on the conclusions to emphasize the differences in clinical outcomes between the two study groups.
R = We thank the Reviewer suggestion. We have corrected this information throughout the manuscript accordingly and emphasize the differences in clinical outcomes between the two study groups in the conclusion:
Round 2
Reviewer 2 Report
No further concern.